# BDS-GCN: Efficient Full-Graph Training of Graph Convolutional Nets with Partition-Parallelism and Boundary Sampling

## Abstract

Graph Convolutional Networks (GCNs) have emerged as the state-of-the-art model for graph-based learning tasks. However, it is still challenging to train GCNs at scale, limiting their applications to real-world large graphs and hindering the exploration of deeper and more sophisticated GCN architectures. While it can be natural to leverage graph partition and distributed training for tackling this challenge, this direction has only been slightly touched on previously due to the unique challenge posed by the GCN structures, especially the excessive amount of boundary nodes in each partitioned subgraph, which can easily explode the required memory and communications for distributed training of GCNs. To this end, we propose BDS-GCN, a method that adopts unbiased boundary sampling strategy to enable efficient and scalable distributed GCN training while maintaining the full-graph accuracy. Empirical evaluations and ablation studies validate the effectiveness of the proposed BDS-GCN, e.g., boosting the throughput by up-to 500% and reducing the memory usage by up-to 58% for distributed GCN training, while achieving the same accuracy, as compared with the state-of-the-art methods. We believe our BDS-GCN would open up a new paradigm for enabling GCN training at scale. All code will be released publicly upon acceptance.

## 1 Introduction

Graph convolutional networks (GCNs) (Kipf & Welling, 2016) have gained increasing attention as they recently demonstrated the state-of-the-art performance in a number of graph-based learning tasks, including node classification (Kipf & Welling, 2016), link prediction (Zhang & Chen, 2018), graph classification (Xu et al., 2018), and recommendation systems (Ying et al., 2018). The excellent performance of GCNs is attributed to their unrestricted and irregular neighborhood connectivity which provides them greater applicability to graph-based data than convolutional neural networks (CNNs) that adopt a fixed regular neighborhood structure. Specifically, given a node in a graph, a GCN first *aggregate*s the features of its neighbor nodes, and then transforms the aggregated feature through (hierarchical) feed-forward propagation to *update* the given node feature. The two major operations, i.e., *neighbor aggregate* and *update* of node features, enables GCNs to take advantage of the graph structure and outperform their structure-unaware alternatives.

Despite their promising performance, training GCNs has been very challenging, limiting their application to large real-world graphs and hindering the exploration of deeper and more sophisticated GCN architectures. This is because as the graph size grows, the sheer number of node features and the large adjacency matrix can easily explode the required memory and communications. To tackle this challenge, several sampling-based methods have been developed for reducing the memory requirement at a cost of approximation errors. For example, GraphSAGE (Hamilton et al., 2017) and others (Chen et al., 2017; Huang et al., 2018) reduce the full-batch of a large graph into a mini-batch via neighbor sampling; alternative methods (Chiang et al., 2019; Zeng et al., 2019) use sub-graph sampling to extract induced sub-graphs as training samples.

In parallel with sampling-based methods, a recently emerged and promising direction for handling large graph training is the distributed training of GCNs, which aims to train large full-graphs over multiple GPUs without degrading the accuracy. The key idea is to partition a giant graph into small

subgraphs such that each could be fit into a GPU, and train them in parallel with necessary communication. Following this paradigm, pioneering efforts, including NeuGraph (Ma et al., 2019), ROC (Jia et al., 2020), and CAGNET (Tripathy et al., 2020), demonstrate the great potential of distributed GCN training, but with different trade-offs. NeuGraph and ROC store entire (sub)graphs in CPU for overcoming the hurdle of still severe requirement of GPU memory, which yet relies on heavy GPU-CPU communications. CAGNET splits the feature vector of nodes into small sub-vectors to reduce the granularity of computation for potential memory saving, which however requires repeated and redundant broadcast of all node features across all subgraphs. As a result, these methods not only require either extra CPU resources or communication traffic, but also hurt training performance.

To enable efficient full-graph training of GCNs without these aforementioned issues, this work sets out to understand the underlying cause of the memory and communication explosion in distributed GCNs training by carefully analyzing the training paradigm – *partition parallelism*. We find that even with *partition parallelism* GCN training can still be ineffective if not designed properly, which motivates us to make the following contributions:

- We identify and formalize two main challenges in *partition parallel* training of GCNs: prohibitive memory requirement and communication volume. We further identify the cause of these challenges to be excessive number of *boundary nodes* within each partitioned graph and such cause is unique in GCNs architecture due to *neighbor aggregation*. These findings provide researchers better understanding in distributed GCN trainings and potentially inspires further ideas in this direction.

- We propose BDS-GCN, a simple yet effective boundary sampling method for overcoming both challenges above, which enables more scalable and performant large-graph training of GCNs while maintaining a full-graph accuracy. BDS-GCN randomly samples features of *boundary nodes* during each training epoch, aggressively shrinking the required memory and communication volume without compromising the accuracy.

- Experiments and ablation studies consistently validate the effectiveness of the proposed BDS-GCN in terms of training performance and achieved accuracy, e.g.,boosting the throughput by up-to 500% and reducing the memory usage by up-to 58% while achieving the same or even better accuracy, as compared with the state-of-the-art methods when being applied to Reddit and ogbn-products datasets.

## 2 BACKGROUND AND RELATED WORKS

**Graph Convolutional Networks.** A GCN takes graph-structured data as input and learn a feature (embedding) vector representing each node in the graph. To learn the feature vector, GCN performs two major steps in each layer, i.e., *neighbor aggregate* and *update*, which can be represented as:

$$a_v^{(l)} = \zeta^{(l)} \left( h_u^{(l-1)} \mid u \in \mathcal{N}(v) \right) \tag{1}$$

$$h_v^{(l)} = \phi^{(l)} \left( a_v^{(l)}, h_v^{(l-1)} \right) \tag{2}$$

where $\mathcal{N}(v)$ denotes the neighbor set of node $v$ in the graph and $h_u^{(l)}$ denotes the learned feature vector of node $u$ at the $l$-th layer. $\zeta^{(l)}$ is aggregation function that takes neighbor features to generate aggregation result $a_v^{(l)}$ for node $v$. Then $\phi^{(l)}$ gets the feature of node $v$ updated. A famous instance of GCN is GraphSAGE with mean aggregator (Hamilton et al., 2017), in which $\zeta^{(l)}$ is mean function and $\phi^{(l)}$ is $\sigma \left( W \cdot \text{CONCAT} \left( a_v^{(l)}, h_v^{(l-1)} \right) \right)$, where $W$ is the weight matrix and $\sigma$ is a non-linear activation. This instance is the focus of our work, but our approach can also be extended easily to other popular aggregators and update functions.

**Large Graph Training.** Real-world graphs consist of millions of nodes and billions of edges (Hu et al., 2020), which are beyond the capability of vanilla GCNs (Hamilton et al., 2017; Jia et al., 2020), especially due to the constraint of GPU memory capacity. To tackle this issue, several sampling-based methods were proposed, such as neighbor sampling (Hamilton et al., 2017; Chen et al., 2017), layer sampling (Chen et al., 2018; Huang et al., 2018; Zou et al., 2019), and subgraph sampling (Chiang et al., 2019; Zeng et al., 2019; Wang et al., 2019). However, these methods suffer three major drawbacks:

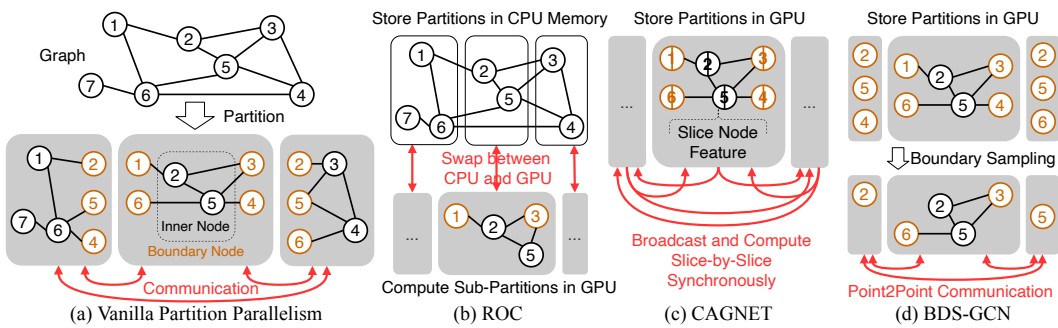

Figure 1: Comparison among different approaches of distributed GCN training for large graphs.

- *Inaccurate feature estimation*: although most sampling methods provide unbiased estimation of node features, the variance of these estimation hurts the model accuracy. As Chen et al. (2017) shows, sampling-based methods with a smaller variance is beneficial to improving the accuracy.

- *Neighbor explosion*: Hamilton et al. (2017) first uses node sampling that randomly selects several neighbors in the previous layer, but as the model getting deeper the size of selected nodes exponentially increases. Chen et al. (2017) and Huang et al. (2018) further propose samplers for restricting the size of neighbor expansion, which yet suffers from heavy memory requirements.

- *Extra time cost for sampling batches*: All sampling based methods require to take some extra time for generating mini-batches, which hurts the efficiency of model training.

**Distributed Training for GCNs.** To train GCNs for large graphs without limitations, distributed training rises as a promising solution – leveraging a cluster of GPUs to enable full-graph training. It is tempting to directly take classical distributed training approaches (such as data and model parallelism) for GCNs. Unfortunately, GCNs training is diverged from the setting of those classical approaches where data samples are small yet the model is large (e.g, model parallelism) and data samples do not have dependency (e.g., data parallelism), both of which violate the nature of GCN training. A GCN-oriented method should be: partition the full (giant) graph into small sub-graphs such that each could be fit into one GPU memory, and train them in parallel, where communication across sub-graphs (GPUs) is necessary to exchange *boundary* node features to perform *neighbor aggregation* of GCNs, which is called *vanilla partition parallelism* as shown in Fig. 1 (a).

Following this paradigm, several works were proposed recently. ROC (Jia et al., 2020) partitions large graphs but stores all partitions in CPU and swaps a fraction of each partition to compute on GPU (see Fig. 1 (b)). It relies on swaps to overfit the GPU memory for large graphs, thus inevitably demanding heavy swap communications, which not only require extra CPU resources and traffic but also hurt the performance. Similar swap-based works are NeuGraph (Ma et al., 2019) and AliGraph (Zhu et al., 2019). CAGNET (Tripathy et al., 2020) also partitions graphs but further splits node feature vectors into small sub-vectors such that communication and compute are reduced to a smaller granularity for memory savings (see Fig. 1 (c)). However, this strategy requires broadcast of each sub-vector for every node in a fully sequential manner, which could incur both redundant communications and an excessive synchronization overhead.

**Distributed Graph Systems.** Distributed graph systems were proposed for analyzing large graphs to solve general graph problems (Shun & Blelloch, 2013; Nguyen et al., 2013; Gonzalez et al., 2012; Zhu et al., 2016; Chen et al., 2019). Lerer et al. (2019) also proposes a distributed learning system for graph embedding. However, none considers node features and cannot be used to for GCN training.

## 3   PARTITION-PARALLEL GCN TRAINING WITH BOUNDARY SAMPLING

**Overview.** To address the aforementioned challenges (i.e., (1) accuracy loss incurred by existing sampling-based methods and (2) heavy swaps and redundant communications imposed by existing distributed training methods), we propose a partition-parallel training of GCNs with a simple yet effective boundary sampling, dubbed BDS-GCN, as shown in Fig. 1 (d). Specifically, BDS-GCN partitions a given full-graph with minimized *boundary node* set and further samples them to shrink both the required memory storage and communication costs in each partition/subgraph, enabling

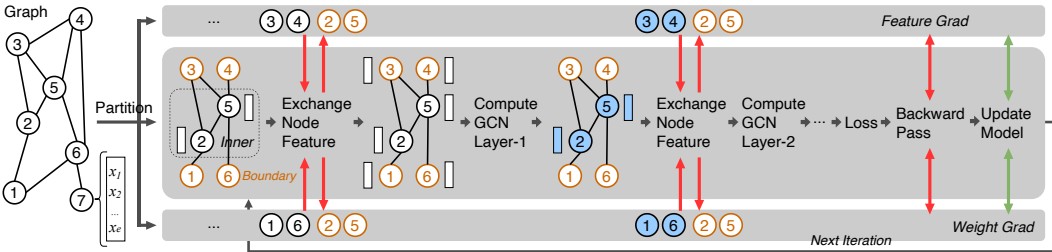

Figure 2: Illustrating vanilla **partition parallel** GCN training, where large graphs are partitioned into smaller sub-graphs (see inner nodes) with each can be fit into one GPU memory. But excessive boundary nodes (in orange) are introduced in each sub-graph by GCNs' *neighbor aggregation*, which not only drastically increases memory cost of each subgraph but also incurs heavy communication overhead between subgraphs, rendering such partition parallelism ineffective.

efficient and scalable large-graph training of GCNs on memory-constrained accelerators without necessitating neither extra host resources nor communication traffics. Notably, BDS-GCN's advantage in substantially reducing hardware costs for training large graphs does not compromise the accuracy, i.e., **BDS-GCN maintains the full-graph accuracy**.

We develop BDS-GCN by first analyzing and identifying the two major challenges (i.e., large memory and communication requirements; see Fig. 2) in existing *partition-parallel* training of GCNs, and then pinpoint the underlying cause to be the excessive number of *boundary nodes* inherited from GCNs' unique *neighbor aggregate* operator. To tackle this cause directly, we design an unbiased random sampling method that solves both above challenges.

### 3.1 CHALLENGES IN VANILLA PARTITION-PARALLEL TRAINING

To enable full-graph training, the original large graph can be partitioned into smaller subgraphs such that each subgraph can potentially be fit into one GPU memory, and then all subgraphs are trained in parallel, i.e., performing GCN compute locally on each subgraph **while also communicating dependent node features across subgraphs**, which we call *partition parallelism*.

As illustrating in Fig. 2, each subgraph in *partition parallelism* contains a subset of nodes in the original graph, which can be called *inner node set* (shown in the dash block) and is unique for each subgraph. In addition, each subgraph is also required to hold a *boundary node set* (shown in orange) which contains dependent nodes from other subgraphs. Such a *boundary node set* is dictated by the nature of GCNs – *neighbor aggregate* of node features that can span multiple neighbor subgraphs. For instance, the node-5 requires the features of nodes-[3,4,6] residing on other subgraphs to perform Equ. 1, thus creating the *boundary nodes* associated to the subgraph hosting node-5. To compute each GCN layer, **features of *boundary nodes* are communicated, i.e., exchanged, across subgraphs** (shown in red) before the features of *inner nodes* get updated (e.g., nodes-[2,5] shown in blue). These updated features are again exchanged across subgraphs to compute the next GCN layer, so on and so forth, until the final layer. The backward pass processes inversely as the forward pass, except that it **communicates the gradients of *boundary nodes*** instead of their features. Then GCN model get updated by aggregating weight gradients (e.g, via allreduce) among all partitions. Note that common GCN models are relatively small compared to the features in subgraphs, rendering their hardware cost negligible (Jia et al., 2020).

Although it seems such a vanilla *partition parallelism* can be used to handle large graph training of GCNs, we find that it neither ineffective nor scalable due to the following challenges:

- Challenge 1: Even with partition, each subgraph can still require prohibitive memory budget to hold both the inner and boundary sets, which can easily overflow GPU memory capacity.

- Challenge 2: The partition parallelism can suffer from severe communication overhead, slowing down the training and limiting its scalability for adopting more partitions/GPUs.

We further identify that both the above challenges share the same underlying cause – **the overhead of extra *boundary node* within each partition**, as elaborated below:

Table 1: Comparison between number of boundary and inner nodes in partitioned Reddit graph with METIS.

| Partition Index | 1 | 2 | 3 | 4 | 5 | 6 | 7 | 8 | 9 | 10 |
|---|---|---|---|---|---|---|---|---|---|---|
| # Inner Nodes | 14k | 15k | 15k | 15k | 15k | 15k | 14k | 15k | 14k | 15k |
| # Boundary Nodes | 39k | 15k | 86k | 78k | 86k | 62k | 6k | 46k | 71k | 23k |
| Ratio of # Boundary to # Inner | **2.64** | 1.00 | **5.45** | **4.95** | **5.49** | **4.11** | 0.42 | **3.04** | **4.81** | 1.52 |

- Each *boundary node* requires **both** storage and communication for calculating **every** layer of GCNs and for **both** forward and backward pass (as illustrated in Fig. 2).

- The *boundary nodes* can have duplication across subgraph.

- The *boundary node set* can be much larger than the *inner node set* in each subgraph.

As shown in Tab. 1, the number of *boundary nodes* in each subgraph can be $5.5\times$ of the number of *inner nodes* for the real-world Reddit (Hamilton et al., 2017) graph partitioned using the METIS algorithm (Karypis & Kumar, 1998), leading to both memory and communication overhead. Analytically, we formalize both cost caused by *boundary nodes* in *partition parallelism* as below:

**Analysis of Memory Cost.** For the $l$-th layer, suppose the input feature is of dimension $e^{(l)}$, and the numbers of inner nodes and boundary nodes in partition/subgraph $P_i$ are $n_{in}^{(i)}$ and $n_{bd}^{(i)}$, respectively. Considering general cases where we save all node features and inner nodes' aggregated features for supporting the back propagation of both Equ. 1 and Equ. 2. When using a GraphSAGE layer with mean aggregator, the memory cost is as follows:

$$\text{Mem}^{(l)}(P_i) = (3n_{in}^{(i)} + n_{bd}^{(i)})e^{(l)} \tag{3}$$

As a result, the number of boundary nodes increases the memory requirement *linearly*.

**Analysis of Communication Cost.** For the partition $P_i$, the communication volume can be defined as $\text{Vol}(P_i) = \sum_{v \in P_i} D(v)$ where $D(v)$ denotes the number of different partitions in which $v$ has a neighbor node, excluding $P_i$ (Buluç et al., 2016). This value quantifies the total amount of features $P_i$ needs to send during each propagation. As the total number of received messages equals to the total number of sent messages, the total communication volume *equals* to the total boundary nodes:

$$\sum_i \text{Vol}(P_i) = \sum_i n_{bd}^{(i)} \tag{4}$$

## 3.2 THE PROPOSED BDS-GCN TECHNIQUE

### 3.2.1 BDS-GCN: GRAPH PARTITION

As boundary nodes are the cause for ineffective *partition parallelism*, the graph partition has to minimize all *boundary node sets* to prevent subsequent memory and communication overheads, dubbed Goal-1. Furthermore, the graph partition must also achieve balanced computation time across all partitions, dubbed Goal-2, since the *partition parallelism* is a synchronous paradigm that requires frequent synchronization at layer granularity (again due to *neighbor aggregate* nature of GCN), where unbalanced partition can result in stragglers blocking other partitions to train.

Many existing graph partition methods aim to achieve the aforementioned Goal-2 yet ignoring Goal-1. In this work, we adopt METIS (Karypis & Kumar, 1998) to implement BDS-GCN. Considering both Goal-1 and Goal-2, we first approximate the compute complexity to determine the size of partitions (e.g., when Equ. 2 is dominated by matrix multiplication, the complexity is proportional to the number of nodes, so we set partitions with equal size in this case), aiming to balance computations across all partitions. After that, we set the objective of METIS to minimize the total communication volumes, which is to minimize the size of *boundary node sets*, according to Equ 4.

### 3.2.2 BDS-GCN: BOUNDARY SAMPLING

Even with optimal graph partition, the *boundary node* issue still remains (see Tab. 1), calling for innovative methods to trim down the *boundary nodes* for enabling efficient and scalable *partition parallelism*. Ideally, such method should achieve all following goals: 1) substantially shrinking

---

**Algorithm 1:** Boundary sampling for partition-parallel training (per-partition view)

---

**Input:** graph partition $\mathcal{G}$, boundary node set $\mathcal{N}$, node features $X$, labels $Y$, sampling rate $p$,
        initial model $w[0]$, learning rate $\gamma$, number of partitions $m$
**Output:** trained model $w[T]$

1   $\mathcal{V} \leftarrow \{\text{node } v \in \mathcal{G} : v \notin \mathcal{N}\}$; // create inner node set
2   $H^{(0)} \leftarrow X$; // initialize input features
3   **for** $t \leftarrow 1 : T$ **do**
4      $\mathcal{N}_s \leftarrow$ randomly select elements in $\mathcal{N}$ with probability $p$;
5      Broadcast $\mathcal{N}_s$ and Receive $[\mathcal{U}_1, \cdots, \mathcal{U}_m]$; // notify selection to all
6      $[\mathcal{V}_1, \cdots, \mathcal{V}_m] \leftarrow [\mathcal{U}_1 \cap \mathcal{V}, \cdots, \mathcal{U}_m \cap \mathcal{V}]$; // record selection of others
7      $\mathcal{G}_s \leftarrow$ node induced subgraph of $\mathcal{G}$ from $\mathcal{V} \cup \mathcal{N}_s$;
8      **for** $l \leftarrow 1 : L$ **do**
9          Send $[H_{\mathcal{V}_1}^{(l-1)}, \cdots, H_{\mathcal{V}_m}^{(l-1)}]$ to partition $[1, \cdots, m]$ and Receive $H_{\mathcal{N}_s}^{(l-1)}$;
10        $H^{(l)} \leftarrow GCN^{(l)}(\mathcal{G}_s, concat(H^{(l-1)}, H_{\mathcal{N}_s}^{(l-1)}))$;
11      **end**
12      $f_{\mathcal{G}} \leftarrow \sum_{v \in \mathcal{V}} \ell(h_v^{(L)}, y_v)$; // calculate loss
13      $g_{\text{local}}[t] \leftarrow \frac{\partial f_{\mathcal{G}}}{\partial w[t]}$; // backward pass
14      $g[t] \leftarrow AllReduce(g_{\text{local}}[t])$;
15      $w[t] \leftarrow w[t-1] - \gamma \cdot g[t]$;
16 **end**
17 **return** $w[T]$

---

*boundary set*, 2) incurring only minimal overhead, and 3) maintaining full-graph accuracy of GCNs. As such, we propose an unbiased random sampling method called *boundary sampling*. (The phrase 'unbiased' refers to the mean aggregator and ignores activations, as in Chen et al. (2017) and Zeng et al. (2019).) The key idea is to select a subset of *boundary nodes* independently from each partition, then **to store and communicate merely those selected *boundary nodes* instead of the full set**, and to update the selection randomly for every epoch.

Algorithm 1 outlines our *boundary sampling* method for *partition parallel* training of GCNs. In each partition, we randomly keep *boundary nodes* with a probability of $p$, and drop the rest at the beginning of each epoch. Then these selected node indices are shared across all partitions such that each partition can "know" others' selections and can record its local node $\mathcal{V}_i$ that is selected by the other $i$-th partition. **During forward pass** of every layer, each partition sends those features $H_{\mathcal{V}_i}^l$ of previously recorded nodes to the corresponding $i$-th partition, respectively, and meanwhile receives features of its own selected boundary nodes to perform GCN operations. (Here, if mean aggregator is used, we replace sent/received feature vector $h$ with $h/p$ for unbiased feature estimation.) **During backward pass** of every layer, each partition sends and receives feature gradients of the selected boundary nodes while generating weight gradients of the GCN model. Lastly, weight gradients are aggregated across all partitions via AllReduce to perform local weight updates.

The proposed *boundary sampling* reduces the number of *boundary nodes* by a factor of $\frac{1}{p}$, together with the same factor of reduction in memory and communication costs (according to both Equ 3 and 4). Meanwhile, *boundary sampling* introduces only negligible overhead due to random sampling on node indices, making it a simple-yet-effective approach. Note that our *boundary sampling* can not only boost the efficiency and scalability of vanilla *partition parallelism*, but also be easily plugged into any *partition parallel* training methods (such as ROC (Jia et al., 2020) and CAGNET (Tripathy et al., 2020)) for furthering their training efficiency.

Lastly, we compare the proposed *boundary sampling* with existing sampling methods:

- Node Sampling: GraphSAGE (Hamilton et al., 2017), VR-GCN (Chen et al., 2017), AS-GCN (Huang et al., 2018) propose node sampling, which would sample **the same nodes** multiple times from the previous layers, which restricts the depth of GCNs and training efficiency. Furthermore, BDS-GCN does not sample neighbors in each subgragh, reducing estimation variance.

- Layer Sampling: *Boundary sampling* is similar to layer sampling as nodes in the same subgraph share the same sampled boundary nodes in the previous layer. Different from layer sampling such as FastGCN (Chen et al., 2018), *boundary sampling* has much denser sampled layers, and thus potentially improves the training quality.

- Subgraph Sampling: *Boundary sampling* can be viewed as one kind of subgraph sampling that drops boundary nodes from other subgraphs/partitions. ClusterGCN (Chiang et al., 2019) and GraphSAINT (Zeng et al., 2019) propose subgraph sampling, yet their number of selected samples are small (e.g., ClusterGCN and GraphSAINT sample only 1.3% and 5.3% of nodes, respectively), leading to a higher variance of their feature estimation.

- DropEdge and Dropout: These techniques can be potentially integrated into BDS-GCN. However, applying them directly is not practical as they do not directly reduce boundary nodes. Most experiments in DropEdge drop at most 50% edges (Rong et al., 2019) and the resulting dropped boundary nodes are still too few to support large graph training.

## 4 EXPERIMENTS

**Datasets and Setup**

We evaluate the efficacy of our methods on two standard large-scale datasets, Reddit (Hamilton et al., 2017) and ogbn-products (Hu et al., 2020), and list used GCN models as following:

- Reddit: An inductive classification task. The graph consists of 233K nodes and 114M edges with a feature size of 602. We use a 4-layer model with 256 hidden units and set learning rate as 0.01 for Adam optimizer with 1500 epochs.

- ogbn-products: A transductive classification task. The graph consists of 2.4M nodes and 62M edges with a feature size of 100. We use a 3-layer model with 128 hidden units and set learning rate as 0.001 for Adam optimizer with 500 epochs.

- Yelp: An inductive multi-label classification task. The graph consists of 716K nodes and 7.0M edges with a feature size of 300. We use a 4-layer model (2 GraphSAGE layers and 2 linear layers) with 512 hidden units and set learning rate as 0.001 for Adam optimizer with 3000 epochs.

To ensure the reproducibility and robustness of BDS-GCN, we do not tune but fix hyperparameters for boundary node sampling throughout all experiments.

We implement BDS-GCN in PyTorch (Paszke et al., 2019) and DGL (Wang et al., 2019). All experiments are conducted on a single machine with 10 RTX-2080Ti (11GB).

**Performance Benefit of BDS-GCN**

We evaluate the training performance of BDS-GCN and compare it with the state-of-the-art works, ROC (Jia et al., 2020) and CAGNET (Tripathy et al., 2020), for distributed training of GCNs on full large graphs. The result is shown in Fig. 3, where each partition uses a single GPU for all methods except that CAGNET ($c$=2) requires doubled GPUs. From Fig. 3, we observe that BDS-GCN outperforms other methods consistently across different number of partitions and different *boundary sampling rate $p$*. When $p = 0.01$, BDS-GCN offers a

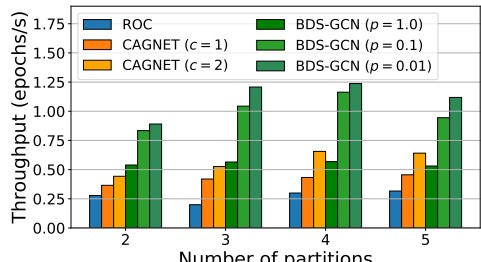

Figure 3: Throughput comparison on Reddit data. Each partition uses one GPU, except CAGNET($c$=2) uses doubled GPUs.

promising throughput improvement of up-to 500% compared with ROC and 118% compared with CAGNET($c$=2). Even when $p = 1$, BDS-GCN still improves the throughput by up-to 190% compared with ROC and 49% compared with CAGNET($c$=1). The advantage of BDS-GCN can be attributed to not only the reduced communication overhead with *boundary sampling*, but also no swap between CPU and GPU (as in ROC) nor redundant broadcast and synchronization overhead (as in CAGNET). Meanwhile, we also find that increasing the number of partitions do not always boost the performance for all methods, because the issue of *boundary nodes* can become more severe and incur more communication overhead.

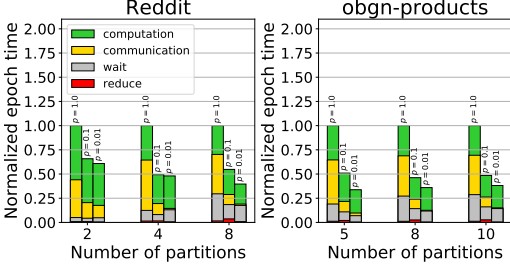 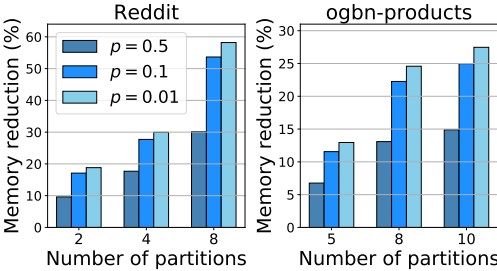

Figure 4: Training time breakdown of BDS-GCN with different *boundary sampling rate*s. Time are normalized against baselines without sampling in each number of partitions.

Figure 5: Application-level memory usage reduction achieved by BDS-GCN. Reduction are against baselines without sampling in each number of partitions.

**Detailed Performance Benefits of BDS-GCN**

To further understand the performance of BDS-GCN, we breakdown the training time into four major components (local computation, communication for boundary nodes, wait, and allreduce on model gradient) and present them in Fig. 4. We observe that local computation time is relatively small but communication (and wait) dominate the total training time, i.e., up-to 52% and 48% of the epoch time are paid for communication in baselines ($p$=1) on Reddit and obgn-products, respectively. With *boundary sampling* ($p < 1$), such heavy overhead is reduced substantially. Especially, the $p$=0.01 almost removes entire communication time for $4 \sim 10$ partitions on Reddit and obgn-products, thus significantly cutting the total training time of GCNs. Note that this experiment is conducted in a single machines where all traffics enjoy high-bandwidth PCIe, but when scaling up further with multiple machines connected via lower bandwidth Ethernet, the distributed GCNs training could suffer from more severe communication bottleneck and thus making the proposed *boundary sampling* more desirable.

Besides, we also examine benefit of BDS-GCN in memory saving. We measure the *application-level* memory usage of BDS-GCN under the same experiment setting, and summarize the achieved memory usage reduction in Fig. 5, where BDS-GCN($p$=1) serves as the baseline. From Fig. 5, we observe that *boundary sampling* consistently reduce memory usage across different number of partition on two datasets. For the denser Reddit, BDS-GCN($p$=0.01) saves 58% memory for 8 GPUs. For the sparser obgn-products, BDS-GCN($p$=0.01) also still saves 27% memory for 10 GPUs.

**Full-Graph Accuracy of BDS-GCN**

Besides improving system performance, BDS-GCN also maintains the accuracy of full-graph training. To validate this, we have conducted extensive experiments to evaluate the test accuracy under the settings of various sampling rates and different numbers of partitions in distributed GCN training, and compare the results with the state-of-the-art sampling-based methods (using GraphSAGE architecture) in Tab. 2 (Hu et al., 2020; with Code, 2020; Chiang et al., 2019; Chen et al., 2017; Zeng et al., 2019; Hamilton et al., 2017; Cong et al., 2020; Zou et al., 2019; Chen et al., 2020). From Tab. 2, we observe that *full-graph training (BDS-GCN with p=1) always achieves a higher accuracy than exisiting sampling-based methods*, regardless of different datasets or number of partitions, which is consistent with results of ROC (Jia et al., 2020).

More importantly, *BDS-GCN always maintains or even increases the full-graph accuracy*, regardless of the sampling rates (e.g., $p$=0.1/0.01), the number of partitions, or different datasets. For instance, on Reddit, *p=0.1 achieves a test accuracy of 97.10%, 97.12%, 97.07% under 2, 4, and 8 partitions, respectively, which are consistently equal or better than the 97.07% of full-graph unsampled training*, thus validating the effectiveness and robustness of boundary node sampling.

However, we also observe that the *special case of BDS-GCN (p=0)* always suffers from the worst test accuracy/score on the three datasets, compared with other cases ($p$=1/0.1/0.01). We understand that the accuracy/score drop is due to the full isolation of each partition after completely removing all boundary nodes, leading to no boundary node features during "neighbor aggregation" throughout the end-to-end training. Therefore, to retain a full-graph prediction accuracy, a relatively high sampling rate like $p$=0.1/0.01 is preferred when using BDS-GCN.

Table 2: Comparison of test accuracy (%) with the state-of-the-art methods on Reddit and ogbn-products, and comparison of test set F1-micro score (%) on Yelp. BDS-GCN with various (*boundary sampling rate*) and under different numbers of partitions are shown. Symbol ¶ denotes the equivalence of full-graph training across a different number of partitions.

| Method | Reddit | | | ogbn-products | | | Yelp | | |
|---|---|---|---|---|---|---|---|---|---|
| Sampling-based methods | | | | | | | | | |
| FastGCN | 93.7 | | | - | | | 26.5 | | |
| GraphSAGE | 95.4 | | | 78.70 | | | 63.4 | | |
| AS-GCN | 96.3 | | | - | | | - | | |
| LADIES | 94.3 | | | - | | | 60.2 | | |
| VR-GCN | 96.3 | | | - | | | 64.0 | | |
| ClusterGCN | 96.6 | | | 78.97 | | | 60.9 | | |
| GraphSAINT | 96.6 | | | 79.08 | | | 64.7 | | |
| BDS-GCN | | | | | | | | | |
| # Partitions | 2 | 4 | 8 | 5 | 8 | 10 | 3 | 6 | 10 |
| BDS-GCN (1.0) | ¶ | 97.07 | ¶ | ¶ | 79.29 | ¶ | ¶ | 64.91 | ¶ |
| BDS-GCN (0.1) | **97.10** | **97.12** | **97.07** | 79.45 | **79.44** | 79.41 | **65.29** | 65.38 | **65.47** |
| BDS-GCN (0.01) | 97.03 | 96.94 | 96.89 | **79.53** | **79.44** | 79.24 | **65.29** | **65.40** | 65.44 |
| BDS-GCN (0.0) | 96.94 | 96.83 | 96.78 | 79.27 | 79.18 | 79.19 | 64.87 | 64.84 | 64.85 |

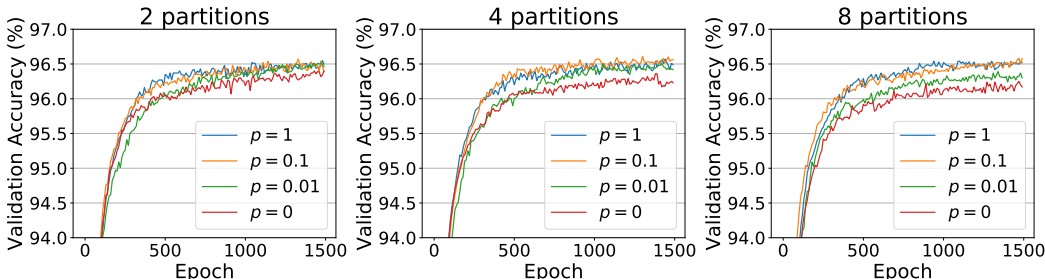

Figure 6: Convergence comparison between unsampled full-graph distributed training (i.e., BDS-GCN with $p$=1) and boundary-node sampled distributed training (i.e., BDS-GCN with $p < 1$) over different numbers of partitions on Reddit.

*To the best of our knowledge, BDS-GCN achieves the best accuracy of training GraphSAGE-layer based GCNs on both Reddit and ogbn-product dataset compared with all existing works.*

**Convergence Speedup of BDS-GCN**

To further understand the affect of BDS-GCN on convergence of distributed GCNs training, we also evaluate the validation accuracy under the same settings of Tab. 2. Fig. 6 compares convergence speed of those settings on Reddit. From Fig. 6, we observe that the boundary node sampling method demonstrates a desirable convergence speed at a high sampling rate. Specifically, *$p$=0.1 achieves the same (or even slightly better) convergence speed as the full-graph training*, regardless of different numbers of partitions. We also notice that (1) *$p$=0 suffers from the slowest convergence (i.e., $0.1 \sim 0.4\%$ drop in validation accuracy from $p$=1)* across different number of partitions, and (2) the convergence gap between $p$=0 and $p$=1/0.1 worsens as more partitions are involved.

## 5 CONCLUSION

In this paper, partition parallel training of GCNs on large full-graph is studied, and its major challenges and underlying cause are identified. With careful analysis of the partition parallelism, a boundary-sampling based method, BDS-GCNs, is proposed and its effectiveness is validated by empirical evaluations. These findings and the proposed method can provide researchers better understanding in distributed GCN trainings and potentially inspires further ideas in this direction.

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
