# OpenReview forum: "BDS-GCN: Efficient Full-Graph Training of Graph Convolutional Nets with Partition-Parallelism and Boundary Sampling"
_ICLR.cc/2021/Conference — Reject_

### Official Review · AnonReviewer3 · 2020-10-27
**Simple and effective idea, lacks novelty**

**Rating:** 4
**Confidence:** 5

**Review:**

-------------
Summary
-------------
This paper presents a system design method to train GNNs on large graphs in a distributed manner. Due to the irregular edge connectivity and the limited GPU memory, intelligent graph partitioning is necessary to support full-batch computation. The authors identify the challenges in computation and communication due to inter-partition connections. Then with the objective of minimizing the edge-cut and improving load balance, the authors use the existing METIS partitioning algorithm to obtain node assignments. To further reduce the amount of communication across GPUs, the author further apply random sampling on the boundary (i.e., inter-partition) nodes. The sampling rate is considered as a hyperparameter to control the tradeoff between accuracy and efficiency. Experiments on two large benchmark graphs show that the proposed method achieves significantly higher hardware efficiency without loss of accuracy, compared with various baselines.

------
Pros
------
+ Clearly defined the motivation and techniques. The overall paper is easy to follow.
+ Strong experimental results (see also cons below for experiments)
+ Distributed GPU training is an important topic

-------
Cons
-------
- Limited novelty: the main algorithmic steps, partitioning and sampling, seems straight-forward. Specifically, the partitioning is based on the well-known METIS algorithm (However, METIS may not be scalable by itself). The "boundary sampling" simply performs IID sampling on the inter-partitioning nodes. This is a simple extension to the random layer sampler of the GraphSAGE minibatch method.
- Motivation for full-graph training: Limited GPU memory is not the only reason for minibatch GNN training. Compared with full-graph training, it is also likely that minibatch training can improve the generalization of the model and thus achieves higher accuracy (as can be verified by the GraphSAGE vs. GraphSAINT rows in Table 2). Therefore, how to justify that full-graph training is important?
- Insufficient experiments: The authors only evaluate the proposed method on two graphs, Reddit and ogbn-product. More importantly, it is not justified why we even need distributed training for Reddit in the first place. The training graph of Reddit can fit in the memory of a single GPU. What is the speedup obtained compared with single GPU training of Reddit? In addition, it seems that the accuracy is not significantly affected with very small p (Table 2). This implies that maybe we don't need boundary sampling at all, and can drop the boundary nodes entirely.
- Unclear experiment setup: the GNN architecture details and the hyperparameter tuning procedures are completely missing. This makes the Table 2 results less convincing.
- Where does the accuracy gain come from? Due to the lack of GNN architecture details, it is hard to judge where the accuracy gain comes from in Table 2. Supposedly, the "GraphSAGE" row corresponds to the case of full-batch training of the GraphSAGE architecture. According to the claim of the paper that dropping neighbors incurs information loss, the BDS-GCN accuracy should achieve at most the accuracy of GraphSAGE. Why the accuracies are higher?

-----------------------------------
Recommendation: Reject
-----------------------------------
In summary, I think this paper lacks motivation and novelty. The experiments are not thorough enough.


--------------
Questions
--------------

1. There is a claim in page 3 that all the sampling based methods have "extra time cost for sampling batches". How significant is such overhead? It seems that the sampling of GraphSAGE and GraphSAINT are pretty cheap -- they are just based on random neighbor selection and random walks. For Cluster-GCN, it seems that the overhead is the same as the proposed BDS-GCN, since both use METIS.

2. Claim on unbiased sampling: Such claim should be more carefully phrased. Under what condition is the sampling unbiased? Only for GraphSAGE-mean? Ignoring the activations?

3. In Figure 4, it seems that increasing the number of GPUs does not give any benefit in terms of training time. What about the time for a single GPU training without any partitioning?

4. In Figure 5, the reduction in memory seems inconsistent with the measurement in Table 1. I would expect that reducing p from 1 to 0 would reduce the memory by multiple folds (since the number of boundary nodes is an order of magnitude larger than the number of inner nodes). However, in Figure 5, the memory is reduced only twice. Why is it the case?

5. What is the GNN architecture and what is the hyperparameter tuning procedure for Table 2?

6. Where does the accuracy gain come from compared with GraphSAGE row in Table 2 (see above "Cons" paragraph)?

7. What is the accuracy if we completely drop the boundary nodes (i.e., setting p=0)?

---

> ### Author Response · Authors · 2020-11-25
> **Response to Reviewer-3**
>
> Thank you so much for your thoughtful review! We present our responses below.
> #### **1. Scalability and objective of METIS**
> Good point. We believe that METIS is scalable enough for current GNN research, as a recent paper from DGL team (https://arxiv.org/pdf/2010.05337.pdf) also uses METIS for partitioning a giant graph of _**100 million nodes**_, which is the largest among all existing GNN datasets.
>
> The METIS objective in BDS-GCN is to minimize the number of _boundary nodes_, instead of edge-cut, because we identified the underlying cause of memory and communication issues in distributed GCNs training to be _boundary nodes_, instead of edges. More details can be found in Sec. 3.2.1.
> #### **2. Limited novelty**
> Thanks for pointing out. Please see question 1 in "Responses to Common Questions".
> #### **3. Motivation for full-graph training**
> According to the recent work [ROC](https://www-cs.stanford.edu/people/matei/papers/2020/mlsys_roc.pdf), full-graph training achieves a higher accuracy than sampling-based methods, which is consistent with our results, i.e., unsampled full-graph training (BDS-GCN with p=1) achieves the highest accuracy among all existing sampled-based methods such as GraphSAINT. Details can be found in our Table 2.
> #### **4. More graphs**
> Good suggestion! To our best effort within a limited time, we have extended our experiments to a third graph as requested, and added the result in Table 2. From it, we observe that the advantage of BDS-GCN still holds on training Yelp with a winning F1-micro score of _65.47%_ vs _64.7%_ of the state-of-the-art GraphSAINT.
> #### **5. p=0, Scalability of GPUs, speedup over one GPU**
> Good points! Please see question 3 and 4 in "Responses to Common Questions".
> #### **6. Unclear experiment setup**
> Sorry for confusion. Experiment setups including model details can be found in “Datasets and Setup” part of Sec 4. For hyperparameter tuning, we do not tune but _**fix**_ hyper parameters for boundary node sampling throughout all experiments. All results reported are based on averaged results of at least five runs. Lastly, _all code will be released upon acceptance to ensure the reproducibility of this work._
> #### **7. Higher accuracy than GraphSAGE**
> Sorry for the confusion. In fact, GraphSAGE in Table 2 is still a sampling-based method to align with the original paper. When comparing the two sampling-based methods, i.e., GraphSAGE vs. BDS-GCN, the higher accuracy of BDS-GCN comes from its advantage of avoiding information loss by only sampling the boundary nodes while always keeping all the inner nodes and their connections.
> #### **8. Overhead of sampling based methods**
> Great point! GraphSAINT has evaluated the overhead of various sampling methods including node/edge sampling, random walks, and multi-dimensional random walks. As reported in Fig. 7 of their appendix, _around **25%** of training time is paid for node, edge, or random walk samplings_ across different datasets, which is a non-trivial cost. The multi-dimensional random walk is even more expensive to execute. By contrast, _the sampler of BDS-GCN takes only around **1%** of training time_. Meanwhile, ClusterGCN also takes much more sampling time than BDS-GCN, because ClusterGCN needs to merge multiple subgraphs into one cluster with a sampling time roughly proportional to the number of edges in the whole graph. By contrast, the sampler of BDS-GCN only needs to modify those boundary edges of selected boundary nodes, and its sampling time is proportional only to the number of boundary edges, which is _only a fraction of ClusterGCN._
> #### **9. Claim on unbiased sampling**
> Sorry for the confusion. Our phrase “unbiased sampling” refers to the _mean aggregator and ignores activations_, just like VR-GCN and GraphSAINT, which has been clarified in the revised manuscript.
> #### **10. Memory reduction in Fig. 5**
> We also observe this issue -- memory reduction is not as expected. However, what Fig. 5 reports is the “application-level” memory reduction, instead of  the “tensor” memory reduction. If we only measure the memory cost by pure “tensors” excluding other objects in the application, then an expected linear reduction in memory is observed in our experiments when scaling down the sampling rate linearly, which matches our Equ (3). Unfortunately, besides tensors, there are other objects also occupying the memory during training, such as buffers/cached blocks of PyTorch, and their memory sizes are not necessarily linear in tensor size. Therefore, even if scaling down the sampling rate, the application-level memory reduction is less than expected. Optimizing memory usage of softwares can get us closer to the expected reduction, but it is beyond the scope of this work.

---

### Official Review · AnonReviewer1 · 2020-10-27
**a simple idea for Full-Graph Training of Graph Convolutional Nets**

**Rating:** 4
**Confidence:** 5

**Review:**

This paper developed a simple method for the distributed training of GCNs. In particular,  the excessive amount of boundary nodes in each partitioned subgraph, which can easily explode the required memory and communications for distributed training of GCNs. To address this problem, this paper proposed BDS-GCN, a method that adopts unbiased boundary sampling strategy to enable efficient and scalable distributed GCN training while maintaining the full-graph accuracy. The experimental results show good performance on large graphs.

1. The writing is good and the idea is clearly presented.

2. The idea is straightforward. It just communicates a subset of neighboring nodes to reduce the communication cost. However, this simple sampling method may incur large variance, slowing down the convergence speed and degenerate the prediction performance.

3. In table 1, why is the number of boundary nodes larger than that of inner nodes?

4. What is the difference between this method and DropEdge (https://arxiv.org/abs/1907.10903)? It seems that this method is identical to DropEdge. In particular, this method drops the edges across different machines. Thus, it is critical to discuss the difference between these two methods, and conduct experiments to compare these two methods.  Otherwise, it is diffcult to see the novelty of this method.

---

> ### Author Response · Authors · 2020-11-25
> **Response to Reviewer-1**
>
> Thank you so much for your thoughtful review! We present our responses below.
> #### **1. A straightforward idea**
> Please see question 1 in "Responses to Common Questions", thanks.
> #### **2. BDS-GCN’s degradation in convergence and accuracy**
> Good point! To understand this, we have conducted extensive experiments to evaluate the convergence and prediction performance of BDS-GCN in various sampling rates and different numbers of partitions. We compare their convergence speeds on Reddit in this figure: [GoogleDrive](https://drive.google.com/file/d/12YPfkW8PveWuXBMPIBMZ5NUA04K27E6l/view?usp=sharing). From this figure, we notice that the boundary node sampling at rate of p=0.1 does not slow down (but slightly increases) the convergence as compared with the unsampled case (p=1), which is consistent across different numbers of partitions.
>
> Furthermore, for the final test accuracy, Table 2 shows that our boundary node sampling always maintains or even increases the full-graph accuracy, regardless of the sampling rates (e.g., p=0.1 or p=0.01) and the number of partitions.
> #### **3. More boundary nodes than inner nodes**
> Each inner node can connect to multiple neighbor nodes residing on other partitions (see illustration in Figure 1(a)) and all those neighbor nodes need to be transferred and aggregated to the inner node (during the “neighbor aggregation” step of GCNs), which makes them the “boundary nodes”. Therefore, each inner node can incur multiple boundary nodes within a partition, and making boundary nodes excessive.
> #### **4. Difference between this method and DropEdge**
> Please see question 2 in "Responses to Common Questions", thanks.

---

### Official Review · AnonReviewer2 · 2020-10-29
**Some baselines are missing.**

**Rating:** 6
**Confidence:** 4

**Review:**

How to perform GCN on large-scale graphs efficiently is crucial in graph learning. Towards this goal, this paper proposes distributed GCN training, BDS-GCN. In particular, it first conducts graph partition and then allocate different partitions into different GPUs.  For each partition,  BDS-GCN samples a portion of boundary nodes (that are shared across partitions) and broacasts the features of the sampled boundary nodes over all partitions. After the message exchanging, BDS-GCN computes one-layer forward pass for each partition  in parallel. The backward pass is done similarly.

In contrast to previous sampling-based methods, this paper can potentially avoid information loss by only sampling the boudery nodes while awalys keeping the inner nodes and their connections.

In contrast to typical distributed graph frameworks, this paper stores each partition locally in each gpu, and the communitation time can be reduced by boundery sampling.

Overall, this paper is good written, the idea is valuable, and the exerimental results generally support the claims the authors have proposed.  Below are some concerns for the current version:

1. It seems in Table 2 that BS-GCN can still obtain promising results when p is small. The reviewer wonders what will happen if p=0, when no boundery node is sampled. This is a crucial baseline that should be compared, as it can tell how important the message exchanging between partitions is, and thus what the boundery sampling stands for.

2. The authors claim that DropEdge is not practical in distributed graph learning, which seems problematic. The communication cost indeed depends on the number of edges between two different partitions. In this sense, it will be interesting if we conduct boundery edge sampling like DropEdge to save communication time. Different from boundery node sampling, boundery edge sampling will keep all nodes (including the boundery nodes) and all connections (including the ones between inner nodes and boundery nodes) within each partition to preserve the information as much as possible, and only those edges connecting different partitions are randomly dropped. It is expected to explore this kind of baseline, and compare it with boundery node samling method.

Other minor issues:

3.   Are the weights stored globally? In a parameter server?

4. What do U1, Um mean in Line 5 in Algorithm 1? Algorithm 1 is only under the per-partition view, and including a global view will better help readers to understand the whole framework.

5.  Some previous results on Reddit by FastGCN, AS-GCN, etc are missing in Table 2.

6. In figure 3, it seems increasing the number of partitions does not necessarily reduce the time cost, why? And how does the accuracy change if we use different number of partitions?

---

> ### Author Response · Authors · 2020-11-25
> **Response to Reviewer-2**
>
> Thank you so much for your thoughtful review! We present our responses below.
> #### **1. BDS-GCN “broadcasts” the features of the sampled boundary nodes**
> A minor fix here. We do not broadcast node features over all partitions. Instead, BDS-GCN uses point-to-point communication to transmit sampled node features (see Figure 1(d) and Line-9 in Algorithm 1). As compared with broadcast, point-to-point communication avoids unnecessary and redundant feature transmission. However, in Line-5 of Algorithm 1, we do have a broadcast at the beginning of each epoch. This broadcast transmits only INDICES of sampled/selected boundary nodes in each partition, instead the node feature vectors, which is meant to notify the other partitions which boundary nodes are selected. If other partitions happen to own those selected nodes, then their node features need to be sent (via point-to-point) to the broadcaster during the “neighbor aggregation.” The overhead of such a broadcast notification mechanism is incurred only once per epoch and is less than 1% of the total training time in our evaluation.
> #### **2. What if p=0**
> Great question! Please see question 3 in "Responses to Common Questions", thanks.
> #### **3. Comparison with DropEdge and BoundaryEdgeSampling**
> Great question! Please see question 2 in "Responses to Common Questions", thanks.
> #### **4. Parameter server**
> We use the standard decentralized data parallelism for maintaining the model, where each worker owns a copy of the model and shares gradients using AllReduce. So the parameter server is not used, which avoids the central bottleneck.
> #### **5. Meaning of U1, Um in Line 5 in Algorithm 1**
> Sorry for the confusion. U_i denotes the indices of sampled boundary nodes at the i-th partition, and each partition receives [U1, …, Um] from other partitions via broadcasting of their sampled boundary node indices.
>
> Specifically, at the beginning of each epoch, each partition needs to sample its boundary nodes and drops the rest (see Line-4 in Alg.1). Next, these sampled/selected node indices are shared across all partitions via broadcast (see Line-5 in Alg.1) such that each partition can know others' selections and thus can send its local nodes that are selected by the other partitions (see Line-6 in Alg.1) during the forward and backward pass.
>
> Such a broadcast notification mechanism is required in BDS-GCN, because it is a fully decentralized design without a global coordinator, and BDS’s random selection of the boundary nodes needs to reach an agreement among all partitions before starting sending the boundary nodes.
> #### **6. Missing results by FastGCN, AS-GCN in Table 2**
> Thanks for pointing out. We have added them to Table 2 in our updated manuscript.
> #### **7. Scalability of more partitions**
> Please see question 4 in "Responses to Common Questions", thanks.
> #### **8. Accuracy change with a different number of partitions**
> Great suggestion. We have conducted extensive experiments after submission and provide accuracy over different numbers of partitions in the Table 2 of our updated manuscript. From the new Table 2, we observe that the test accuracy of BDS-GCN retains the same level across different numbers of partitions. For instance, on Reddit, _BDS-GCN with p=0.1 achieves a test accuracy of **97.10%, 97.12%, 97.07%** under 2, 4, and 8 partitions, respectively_, which are consistently equal or better than accuracy of _**97.07%**_ offered by full-graph unsampled training with p=1, thus validating the robustness of boundary node sampling.
>
> Furthermore, we also compare the convergence speed under different numbers of partitions on Reddit here: [GoogleDrive](https://drive.google.com/file/d/12YPfkW8PveWuXBMPIBMZ5NUA04K27E6l/view?usp=sharing). From this figure, we notice that the boundary node sampling at the sampling rate of p=0.1 maintains the same convergence speed as the unsampled case (p=1), which is consistent across different numbers of partitions.

---

### Official Review · AnonReviewer4 · 2020-10-30
**This paper addresses the efficiency issue raised by GNN training on large graphs. The proposed solution falls in the category of data parallelism, which aims to partition graph into smaller parts while reduce the communication cost caused by extensive number of boundary nodes. The proposed idea is interesting and practical, and the experimental results have demonstrated the superiority of the proposed approach.**

**Rating:** 6
**Confidence:** 4

**Review:**

This paper addresses the efficiency issue raised by GNN training on large graphs. The proposed solution falls in the category of data parallelism, which aims to partition graph into smaller parts while reduce the communication cost caused by extensive number of boundary nodes. The proposed idea is interesting and practical, and the experimental results have demonstrated the superiority of the proposed approach.

Strengths:
1.	This paper successfully identifies the issues in the data parallelism-based GNN training algorithms – each graph partition share too many nodes with other partitions, called boundary nodes, which significantly affect memory and communication cost.
2.	The proposed solution is simple yet neat, which is to sample the boundary nodes to reduce memory and communication cost.
3.	The experiments are convincing, boosting the throughput (meaning number of epochs per sec) by up-to 500% and reducing the memory usage by up-to 58%, while achieving the same or even better accuracy.

Weaknesses:
1.	The graph partition idea is interesting, but more details on how to apply METIS is expected.
2.	The experiments can be further enriched.


Detailed comments:
1.	In this paper, it has mentioned many related studies. But in the experiment section, only a small subset of the mentioned methods are treated as baselines. Also, for the data partition-based baselines, only throughput is compared. How about the accuracy comparison? Which method needs more epochs to converge, which will affect the run-time in addition to the throughput? For the sampling-based methods, more methods are expected to be compared to the proposed approach. In addition to accuracy, run-time comparison is also expected.

2.	For the sampling-based approach, one more baseline is expected:

Zou et al., Layer-Dependent Importance Sampling for TrainingDeep and Large Graph Convolutional Networks, NeurIPS 2019.

---

> ### Author Response · Authors · 2020-11-25
> **Response to Reviewer-4**
>
> Thank you so much for your thoughtful review! We present our responses below.
> #### **1. Details of METIS**
> Good suggestion! To use METIS for graph partitioning, one needs to assign a weight to each node so that the algorithm can separate a whole graph into multiple subgraphs with each having a similar total number of node weights. Meanwhile, one also sets the objective of METIS to minimize either the edge-cut or the communication volume. As a result, graph partitions are obtained with balanced node weights while minimizing the edge-cut or communication volume.
>
> To apply METIS to BDS-GCN, we assign the weight of each node to be its computational cost (of calculating its features, activations, and gradients), thus balancing the computation time of each partition under distributed GCN training. Meanwhile, we set the objective of METIS to minimize the communication volume, which is to minimize the number of boundary nodes (according to Equ (4)), thus minimizing both the communication and memory costs (according Equ (3)). Although such a partition method is heuristic, it helps address the root cause of distributed GCN training’s ineffectiveness directly, and has demonstrated desirable performance in all our evaluations.
> #### **2. More experiments**
> Great point! To enrich our experiments, we have conducted extensive evaluations and ablation studies. For experiments of more baselines and under more sampling rates, please see question 2 and 3 in "Responses to Common Questions", thanks.
>
> For more accuracy comparison, please see the revised Table 2 in our updated manuscript.
>
> For more convergence comparison of various settings on Reddit, please see the figure here: [GoogleDrive](https://drive.google.com/file/d/12YPfkW8PveWuXBMPIBMZ5NUA04K27E6l/view?usp=sharing). This figure shows that the boundary node sampling method demonstrates a desirable convergence speed. Specifically, the boundary node sampling at the sampling rate of p=0.1 maintains the same convergence performance as the unsampled case (p=1, i.e., full graphs), and this advantage is consistent across different numbers of partitions.
> #### **3. Baseline: LADIES**
> Thanks for pointing out! We have included this work into our Table 2 as a baseline method.

---

### Author Response · Authors · 2020-11-25
**Responses to Common Questions: Part 1**

We’d like to thank all reviewers for their thorough review and insightful suggestions. To address the common questions they raised, we have centralized our responses below.
#### **1. Straightforward idea, limited novelty**
We agree that our idea is straightforward yet humbly disagree about the limited novelty. In fact, Reviewer 2 and 4 clearly pointed out that our novelty is nontrivial. Furthermore, we clarify our novelties from two aspects:

* To our best knowledge, this is the first work to _(i) provide a clear understanding about why distributed GCNs training is ineffective (i.e., memory and communication explosion), and (ii) identify the underlying cause to be the excessive number of boundary nodes, rather than boundary edges as many works assumed_ (see the next question). These findings provide researchers better understanding about distributed GCN training and can potentially inspire more innovative ideas in this direction.

* With these findings, we propose the boundary node sampling to overcome both memory and communication challenges while maintaining a full-graph accuracy. _To our best knowledge, BDS-GCN achieves the best accuracy of training GraphSAGE-layer based GCNs on both Reddit and ogbn-products dataset compared with existing works_, while largely reducing memory and communication cost.  Given the growing need for training larger graphs and for exploring more complex GCN models, we believe that the benefits of BDS-GCN will become more pronounced while being compatible with other techniques.

#### **2. Comparison with baselines like DropEdge and BoundaryEdgeSampling**
To clarify the difference between BDS and DropEdge/related-baselines, we provide further analysis and extra experimental comparisons below.

First, DropEdge was proposed as a data augmentation technique to alleviate over-fitting or over-smoothing issues of training GCN, especially on small graphs, by randomly dropping graph edges at a global scale, without any awareness of the edges that reside across or within a partition. DropEdge could be applied directly to distributed GCN training, _but the resulting dropped edges might not be around the boundary, which thus doesn’t reduce the communication cost._

Second, as Reviewer-2 suggested, DropEdge can be adapted to drop only the boundary edges, i.e., _“BoundaryEdgeSampling (BES)”_. This method indeed could reduce communication cost but not as directly/much as our BDS. Because, in real-world graphs, _multiple boundary edges across partitions can connect to the same boundary node_. Even if we drop some of those edges, the remaining undroped edges still demand communication of the connected boundary node to satisfy “neighbor aggregation” of GCNs. Obviously, to root out such communication, boundary nodes should be addressed and dropped directly, instead of using boundary edges. Analytically, we show that the communication cost of distributed GCN training is only proportional to the number of boundary nodes (see Equ (4)).

Finally, to justify the advantages of our BDS-GCN, we implemented the suggested “BES” and provide the following comparison under the same setting of Table2.

For Reddit (2 partitions),

| Method                           	| DropEdge 	| BES 	| **BDS-GCN** 	|
|----------------------------------	|----------	|----------------------	|---------	|
| Communication vol (MB) 	| 301.3    	| 207.9                	| **30.4**   	|
| Epoch time (s)                 	| 2.47     	| 1.70                 	| **1.23**    	|
| Test accuracy (%)                	| 97.07    	| 97.11                	| **97.10**   	|

For ogbn-products (5 partitions),

| Method                           	| DropEdge 	| BES 	| **BDS-GCN** 	|
|----------------------------------	|----------	|----------------------	|---------	|
| Communication vol (MB) 	| 989.2    	| 328.0                	| **100.9**   	|
| Epoch time (s)                 	| 3.03     	| 1.52                 	| **1.24**    	|
| Test accuracy (%)                	| 79.40    	| 79.38                	| **79.45**   	|

For a fair comparison, all methods drop the same number of edges over the full graph, i.e., global edge drop rate of 12% for reddit and 5% for ogbn-products. As can be seen, DropEdge requires the most communication volume per epoch and thus the highest training time, because its randomly dropped edges do not necessarily reside around partition boundaries. By contrast, as our reviewers conjecture, BES reduces communication and training time thanks to its focused boundary dropping. As expected, our BDS-GCN achieves double win among these methods with a great margin. To specific, _BDS-GCN cuts communication volumes by up-to around **10x** and **7x** compared with DropEdge and BES, respectively, and thus speeding up training time by up-to **2.4x** and **1.4x**._ This is attributed to the advantage of BDS-GCN of directly addressing the root cause of the large communication cost -- boundary nodes, in distributed GCN training.

---

> ### Author Response · Authors · 2020-11-25
> **Responses to Common Questions: Part 2**
>
> #### **3. Comparison with p=0**
> We evaluate BDS-GCN with p=0 corresponding to settings of Table 2 under different numbers of partitions for a thorough analysis. The new Table 2 can be found in our updated manuscript. As a quick summary, we provide the test accuracy comparison below.
>
> | Sampling rate 	| Reddit (8 partitions) 	| ogbn-products (8 partitions) 	|
> |:-------------:	|:---------------------:	|:----------------------------:	|
> | p=1.0         	| 97.07%                	| 79.29%                       	|
> | p=0.1         	| 97.07%                	| 79.44%                       	|
> | p=0.01        	| 96.89%                	| 79.44%                       	|
> | p=0           	| 96.78%                	| 79.18%                       	|
>
> From the accuracy aspect: the above comparison demonstrates that training with p=0 suffers from a test accuracy drop of _0.29%_ and _0.26%_, on Reddit and ogbn-products, respectively, compared with the corresponding best-case accuracy (p=0.1). We understand that the accuracy drop is due to the complete isolation of each partition after removing all boundary nodes, leading to no information of boundary node features during “neighbor aggregate” throughout the end-to-end training.
>
> From the convergence aspect: we further compare the convergence speed of p=0 with other settings and provide their training curves on Reddit here: [GoogleDrive](https://drive.google.com/file/d/12YPfkW8PveWuXBMPIBMZ5NUA04K27E6l/view?usp=sharing). This figure shows that (1) p=0 suffers from the slowest convergence **_(i.e., 0.1~0.4% drop in validation accuracy from p=1)_** regardless of the number of partitions, and (2) the convergence gap between p=0 and p=1/0.1 worsens as more partitions are involved. By contrast, p=0.1 demonstrates the same (or slightly better) convergence speed as the full-graph training (p=1), validating the effectiveness of boundary node sampling.
>
> Finally, we’d like to conclude our understanding regarding p=0 and BDS-GCN. First, p=0 is a special case of our proposed BDS-GCN that provides a general boundary node sampling method supporting various sampling rates. Second, BDS-GCN promises a new direction for more favourably trading-off the accuracy and cost of distributed GCN training, as compared to existing methods, such as DropEdge.
>
> #### **4. Scalability of more partitions, speedup over one GPU**
> We also observe this issue -- more partitions does not always improve training time for ALL methods (including all baselines and ours). Through Fig. 4, we find that (1) the compute time does decrease with more partitions, but (2) what dominates the total training time is the communication of boundary nodes (and its incurred wait time). Especially, _with more partitions, this communication overhead can increase because the boundary node issue can become more severe_ (see Sec. 3.1). Therefore, boundary node sampling is more necessary when scaling up the number of partitions to enable large graph training. However, even after sampling, the reduced communication cost can still saturate the training time improvement with more partitions, as in Fig. 3. This is because a fixed sampling rate can only shrink the communication cost by a constant ratio, which still leaves an increased communication cost with more partitions. To achieve even better scalability, adaptive sampling rates of BDS-GCN can be employed to further cut communication, which is an interesting direction to explore in our future works.
>
> Regarding the single GPU training on Reddit, it is potentially possible considering the relatively smaller graph of Reddit. However, given our best GPU resources  (RTX GPU with 11GB memory, commonly adopted in academia groups), one GPU cannot hold the entire graph and model (4-layer GraphSAGE model with 256 hidden units), and thus distributed training is still necessary. To enable training Reddit in one GPU, we have to halve the mode size (from 256 to 128 hidden units). As such, we can evaluate the training speedup with BDS-GCN (p=1), as shown in the table below.
>
> | # Partitions (GPUs)      	| 1    	| 2    	| 3    	| 4    	| 5    	|
> |--------------------------	|------	|------	|------	|------	|------	|
> | Normalized epoch time    	| 1.00 	| 0.87 	| 0.72 	| 0.87 	| 0.88 	|
> | Epoch time (s)         	| 1.54 	| 1.34 	| 1.12 	| 1.34 	| 1.36 	|
> | Communication time (s) 	| 0    	| 0.56 	| 0.55 	| 0.96 	| 1.02 	|
> | # Boundary nodes (10^5)  	| 0    	| 1.03 	| 1.79 	| 2.58 	| 2.92 	|
>
> This table shows that increasing GPUs from a single one reduces training time, but further increasing GPUs beyond 4 saturates the improvement (similar to Fig. 3), because of _the dominating communication time and the increasing boundary nodes with more partitions_.

---

### Decision · Program_Chairs · 2021-01-07
**Final Decision**

**Decision:**

Reject

**Comment:**

The paper is concerned with improving the scalability of GCNs which is an important problem and relevant to the ICLR community. For this purpose, the authors propose a new distributed training method for GCNs which uses a boundary sampling strategy to reduce the number of boundary nodes. The paper is written well and, overall, good to follow. Reviewers highlight the promising improvements in throughput and memory footprint as well as and the possibility to avoid potential loss of information compared to other methods.

However, currently there exist still concerns around the manuscript. Reviewers raised concerns with regard to the novelty of the method (straightforward combination of existing techniques) as well as the experimental evaluation (overhead of METIS, full-batch accuracy, edge boundary vs node boundary, etc.) The revised version addresses some concerns (comparison to DropEdge, p=0, etc.) and clearly improves the paper. However, given the aforementioned issues and the absence of strong support from reviewers, I agree to that the current version would require an additional revision to iron out these points. The presented results are indeed promising and I'd encourage the authors to revise and resubmit their work considering the reviewers' feedback.